# The Impact of Interpretive Packaged Food Labels on Consumer Purchase Intention: The Comparative Analysis of Efficacy and Inefficiency of Food Labels

**DOI:** 10.3390/ijerph192215098

**Published:** 2022-11-16

**Authors:** Muhammad Zeeshan Zafar, Xiangjiao Shi, Hailan Yang, Jaffar Abbas, Jiakui Chen

**Affiliations:** 1Department of Business Administration, University of Chakwal, Chakwal 48800, Pakistan; 2Business School, Shandong Jianzhu University, Jinan 250101, China; 3Institute of Business Administration, Shandong University of Finance and Economics, Jinan 250014, China; 4School of Media and Communication & Antai College of Economics and Management, Shanghai Jiao Tong University, Shanghai 200240, China; 5School of Economics and Management (Cooperative College), Qingdao Agricultural University, Qingdao 266109, China

**Keywords:** food label, consumer purchase intention, attitude

## Abstract

The objectives of this study are twofold. Firstly, the current study elucidates the impact and efficacy of food labels in developing consumers’ attitudes and intentions towards the selection of nutritional food. Secondly, the inefficacy of labels in developing consumers’ attitudes and intentions towards healthy packaged food selection is demonstrated. The supportive theories of the current model are those of reasoned action and protection motivation. The data of 797 respondents have been collected from four major grocery stores in Pakistan. The structural equation model has been employed for the analysis of data. The results indicate that the efficacy of food labels has a positive significant effect on attitudes towards familiar and unfamiliar foods. In contrast to this, inefficacy in labelling has shown a positive significant effect on familiar foods but is insignificant for unfamiliar foods. The user-friendly food labels significantly affect unfamiliar foods in terms promoting consumer attitudes. Reciprocally, the inefficacy of labels creates a hindrance to the reading of unfamiliar labels while purchasing food items. The study findings reveal the fact that food label information and its format influences consumer attitudes and intentions at the point of purchase.

## 1. Introduction

Numerous studies have explored public health issues and the reported results have revealed that they often relate to consumers’ irrational behaviors toward the selection of unhealthy food items [1]. The cause of unhealthy food intake is the profound change in individual eating behaviors over the last five decades [2]. Globally, it has been observed that individuals’ food intake habits have been transformed from home-cooked foods to ready-to-eat, packaged food items [3]. The increasing tendency towards packaged food has affected the medical expenses of individuals on a national scale. A worldwide survey has identified the fact that a poor diet is the major cause of increasing medical expenses, not only for individuals but for the national economy [4]. Therefore, researchers have emphasized the need to investigate the effects of food on the lives and well-being of consumers [5,6]. Additionally, scholars have suggested introducing effective and scalable interventions for consumer awareness to encourage the selection of healthy food items [3].

To achieve food system sustainability regarding healthy food consumption, both food and nutrient literacy play pivotal roles [7]. Education regarding nutrients motivates individuals to make healthy food choices [8]. There is no formal education system to create awareness among individuals regarding healthy packaged food besides food label information. Intuitively, food label information is the most effective tool to guide consumers in the food selection process [9]. The objective of food label information is to provide information about product ingredients and in turn allow the consumer to select healthier food items at the point of purchase [10]. Additionally, the nutritional label is a tool that informs the public about healthy foods, protects the consumer from unsafe foods, and discourages food manufacturers from producing unhealthy and defective food items [11]. There is a need to motivate consumers to consult label information at the point of purchase [5,12]. If they are not motivated to consult label information, as previously mentioned studies have found, the impact of food label information is void because the decisions are made without the customer engaging with the available information [13]. Although food label information is the most widely proposed tool for educating consumers at the point of purchase [14], the reported results regarding the usefulness of food labels at the point of purchase are contradictory.

The reading and understanding of food label information requires special proficiency. Therefore, studies have suggested the design of food labels which reduce label complexity, provide simple and meaningful numeric information and simple text, reduce the use of percentages, and use easy-to-understand label presentation [15,16]. The friendly label allows the consumer to read it before putting a food item into their shopping cart. Similarly, the research also highlights factors that cause consumers to avoid the food label information at the point of purchase, e.g., a lack of understanding, lack of usefulness, lack of trustworthiness, and technical label information [17]. These factors lead to the inefficacy of food labels at the point of selection. However, several authors have reported the effectiveness of food labels for raising the awareness of consumers [18]. Owing to convenience, packaged food products attract consumer attention and have gained a substantial market share. It is very difficult to motivate consumers to stop consuming processed foods, and creating awareness in order to reduce the quantity is one of the only options available [19]. A substantial amount of scientific literature has indicated that the consumer selects food items on the basis of taste [20,21]. Reciprocally, a common notion prevails among consumers that healthier food is less palatable [22,23]. Hence, informative and unique food labels are required to attract consumers’ attention when choosing the appropriate food items. Moreover, familiar and unfamiliar food labels influence the behavior of consumers differently [23,24], because familiar food labels can easily attract consumers’ attention, whereas unfamiliar food labels require further effort to have an impact on consumers’ minds, as well as building their trust [25].

The prior studies’ inconsistent results have motivated the current researchers to investigate the impact of food label efficacy and inefficacy in developing individuals’ attitudes towards familiar and unfamiliar food labels. Reciprocally, the objectives of the present study are twofold. The first is to study the efficacy of the food label information on the attitudes of consumers when consulting familiar and unfamiliar food labels to make nutritional food choices. The second is examining the impact of the inefficacy of the food label information on the consumers’ intentions when consulting familiar and unfamiliar food labels for healthy food selection.

## 2. Literature Review and Hypothesis Development

There are multiple causes of early death, and a poor diet is one of them [26]. Although food-processing companies print detailed information regarding food ingredients and nutrients, food companies are, nevertheless, the drivers of obesity and non-communicable diseases [27]. In contrast to this, food labels allow consumers to make appropriate decisions by providing the necessary information regarding healthy and hygienic food products, and they are an essential tool to provide consumers with knowledge about food ingredients [28,29]. The primary source of communication between consumers and organizations is food labeling, which often influences consumers’ purchase decisions [30,31]. Moreover, food label guides consumers in making right choices [32]. The aforementioned studies have revealed that well-written and detailed information at label is necessary for consumer products [33]. The information written at labels makes an individual able to evaluate the characteristics of food products, interpret correct information and later choose products according to their lifestyle, preference, and health condition [34]. It noticed in the past few years that consumers become more conscious regarding their health and selection of food items. Therefore, consumer’s demand for more transparent food label information increased about nutrients, health benefits, and ingredients [35,36].

Furthermore, to promote their products food processing companies utilize the food labels as promotional tool. [15,37]. In this modern era of consumerism, consumers are becoming more alert and conscious about the nutritional value of the food items [28,38]. Consumers’ decision regarding food selection has mainly influenced by some factors like; the quality, packaging, and labeling of the food products [39,40]. Nevertheless, past studies have observed contradictory results regarding the decisiveness of label. There are numerous scholars in the past, who have examined the influence of label fonts, colors, information and formats on consumer perception [41]. Reciprocally, the marketing activities of food processing companies shape consumer’s opinion. Moreover, studies have accounted that individuals are least interested towards label information. The cause of consumer’s least interest towards food label information is lack of understanding, lack of confidence, proficiency about written nutrients, and lifestyle [42,43]. Therefore, it is necessary to examine the factors that motivate and demotivate consumers to consult food label information at the point of purchase [30].

### 2.1. Efficacy of Food Label Effect on Familiar and Unfamiliar Label and Selection Intention

Efficacy of food label schemes is an important tool. It evaluates the nutritional content at the point of purchase [44,45,46]. Besides, the frequent purchase of packaged food items make consumer familiar with label information. Additionally, the food label information needs numeric proficiency. In continuation with, familiarity and efficacy of food label is helpful for healthy food choice [47]. Reciprocally, to understand the information at food label with familiar food label assistant individual in making healthy packaged food intention [48]. The aforementioned studies have unfolded that, when individual intend to purchase unusual packaged food item, the efficacy of food label information play vital role [49]. The easy to read food label information makes individual confident while purchasing new packaged food item. The food label is a promotional tool. The presentation of food label information regarding nutrients, manufacturing and expiry dates and companies basic information develop trust among consumers [50].

Furthermore, food label information is the fundamental means to communicate information to customers about characteristics of food. Hence, the useful attributes of food label information has investigated multiple times [51]. Studies have suggested to examine the effects of familiarity and unfamiliarity of food labeling on various food products [52]. The aforementioned studies have unfolded the fact that the primary concern of researchers regarding food labeling is nutrition composition and the way nutritional information has displayed at food labels [53]. The European Food Information Council has indicated that consumers in the United Kingdom (UK) are habitual to seek for labels’ information before purchasing the food items for both the familiar or unfamiliar food labels. However, in comparison to the UK, the results are inconsistent in the rest of the world regarding the consultation of food labels for healthy food selection [51]. Besides, some formats of food label could not convince individuals in changing their packaged food choices. The cause of avoiding food label is lack of numeracy about food label contents [54]. Nevertheless, food processing companies and retailers have put collective efforts in devising attractive and easy to read labels for the convenience of consumers. Therefore, the current study has examined the efficacy of food labels in influencing consumers’ purchase decisions. Hence, the current study proposed the hypothesis.

**H1.** 
*Efficacy of food label positively affects consumer’s intention and attitude for familiar label mediates in making this relationship significant.*


**H2.** 
*Efficacy of food label positively affects consumer’s intention and attitude for unfamiliar label mediates in making this relationship significant.*


### 2.2. Label Inefficacy Effects on Familiar and Unfamiliar Label Reading Attitude and Selection Intention

It is an unequivocal fact that food label is useful for educating consumer for the healthier product selection. On the other hand the display of nutritional information with scientific terminologies create hindrance for individual to consult the label at point of purchase for healthy food selection [55]. The effect of inefficacy of food label on unfamiliar food label is significant. Hence, consumers tend to believe in the label while purchasing products [56]. More specifically, the decisiveness of food label information becomes more prominent for unfamiliar food label products. Most often individual is intend to add some newly introduced packaged food items in their shopping list, therefore, individual keenly observe the food label which is unfamiliar to them. Therefore, sometimes unfamiliarity is not the cause of individual demotivation [57]. Hence, the nutritional table is quite confusing that cause the avoidance to read unfamiliar food label [58]. The most common reasons to avoid food label information are reading difficulties, complicated comprehension, lack of time, and lack of information search behavior [59]. Reciprocally, studies have witnessed the effect of inefficacy of food label at familiar food label reading is weak as compare to unfamiliar food label [60]. The packaged food items, which are in regular use of consumer, are familiar with the information given at food label. On the other hand, some of the studies have reported that familiar and unfamiliar food labels have some common information and individual read that text and can select the required packaged food item [61]. In contrary to that past literature have revealed that the impact of food label efficacy and inefficacy are vital in using familiar and unfamiliar label reading while choosing right amount of packaged items. Furthermore, studies have also reported that consumer’s exposure towards food labels also depends upon their household size, purchase pattern of household members, and household composition. The verity of products have consumed by different household members and how many multiple food choices exist in one family member can also be the cause of food label familiarity and unfamiliarity [62]. Hence, the current study has hypothesized that:

**H3.** 
*Inefficacy of food label positively affects consumer’s intention and attitude for familiar label mediates in making this relationship significant.*


**H4.** 
*Inefficacy of food label positively affects consumer’s intention and attitude for unfamiliar label mediates in making this relationship significant.*


The intention of an individual towards any objective is the consequences of an attitude [63]. Reciprocally, the attitude is use to explain the change behavior of an individual [64]. Past literature has revealed that in making a strong intention of an individual attitude play vital role [65]. Therefore, consumer moods most often varies while selecting food related products. Hence, there is a dire need to examine that which factors affect individual’s attitude to consult label information while making a decision regarding food items. Moreover, studies have revealed that label information is basic source of guidance for consumers at purchase point [66]. Besides, the reading attitude of food labels has associated with the highest quality of diets [67,68]. Numerous scholars have reported that most of the time individual’s reported figures regarding reading of label information contradicts from actual behavior [69,70]. There is substantial percentage of packaged food items exists in shopping list of consumers. Therefore, due to routine purchase of the packaged food products consumer become familiar with the label information. Hence, to encourage consumer to read familiar food label is an easy task. In contrary to that, to motivate consumer attitude to consult unfamiliar food label is necessary to make healthy and right amount of food selection intention [71]. The nutritional information, which has mentioned at food label, has most often found confusing. Nevertheless, food label has considered a standard for promoting healthy food items among individuals [59]. To empirically test the above assertions, the current study has put forward the following hypotheses.

**H5.** 
*Attitude towards familiar food label significantly effect on nutritional food selection intention.*


**H6.** 
*Attitude towards unfamiliar food label significantly effect on nutritional food selection intention.*


### 2.3. Theoretical Framework

The proposed model has underpinned with reasoned action theory (TRA) and theory of Protection motivation (PMT). Moreover, to investigate the volitional behavior, the reasoned action theory has most often employed. More specifically past scholars have rendered the services of reasoned action to examine the volitional behavior of food consumption, sustainable and ethical consumption, and organic food selection [72]. Reciprocally, the theory of reasoned action predicts consumer probability, purchase intention and a sensible effort for buying any product [73]. Theory of reasoned action assists scholars in examining whether external factors directly effect on individual’s behavioral intention or not [74]. Furthermore, according to theory of reasoned action, intention is the strong predictor of actual behavior and intention of an individual depends on strong attitude. Additionally, TRA investigates that how individual behave with pre-existing attitude [75]. Although TRA investigates individual intention towards any object but the purpose of present study was to examine the motivational features, which build strong intention about selection of nutritional food items.

Therefore, the current study has rendered the services of Protection Motivation Theory [76]. Protection motivation theory seeks the clarity in cognitive process of an individual while taking any decision. According to the definition of PMT “*Protection Motivation Theory (PMT) emphasis on the cognitive processes mediating attitudinal and behavioral change*” [76]. Besides, PMT posits two processes; one is the arousal of threat and second is the coping of threat. These two characteristics of PMT support current authors to achieve the objective of study. The available information at food label states a threatening situation for an individual if it is not useful for making healthy food selection. Similarly, it also states a coping situation for individual if it is easily understandable and makes consumer intention for healthy package food product. Therefore, the effect of efficacy and inefficacy of food label in making individual attitude to consult familiar and unfamiliar food label at point of purchase is very decisive for healthy packaged food selection intention. Moreover, according to the definition of the protection motivation theory the behavior discusses to motivate or demotivate individual with the belief that the use of preventive measure for any behavior reduces the risk factors. Likewise, the efficacy and inefficacy of food label motivate and de-motivate individuals to read familiar and unfamiliar food label. The researchers of the present study have examined that food label develop individual’s attitude to read food label and strong attitude effect on individual’s intention. Additionally, the food label reading attitude move forward in making intention of an individual to consult label information at the time of food item selection. Figure 1 is the graphical representation of proposed model. 

## 3. Methods

The objective of the study is to find the effect of efficacy and inefficacy of food label information in making consumer attitude with familiar and unfamiliar food label. To achieve the objective of the study respondents’ point of purchase opinion is decisive. Besides, authors of the study have intended to explore impact of food labels efficacy on unfamiliar food label information. Reciprocally, studies have reported that consumer’s responses at the time of shopping most often validate their actual behavior [77]. Therefore, for data collection authors of the study selected four retail outlets. The authors of the study have devised selection criteria for the selection of four retail stores such as the daily footfall of customers, verity of packaged food displayed at their shelves, the population of city where theses outlets have situated, the diversity of customers with respect to gender and age and the income level of customers.

The convenience sampling technique has used to collect data from grocery stores. The convenience sampling technique has supported authors for finding relevant respondents who can willingly participate in research survey. Because the results reported in past literature has evidence that most often customers get offended if someone interferes during shopping [78]. Therefore, the convenience sampling technique has used which demonstrate that respondents have shown their consent to participate. Furthermore, authors of the study have tested their objective with empirical data therefore the data collected using a structured questionnaire. It was adapted questionnaire. The detail of instruments has reported in Table 1.

The distribution of questionnaire among respondents is very technical. Therefore, authors of the study have contacted store managers. The detail of study and objective has discussed with managers. As per the guidance of managers, a written request forwarded for final approval. Besides, due to the customer safety policy, store managers have placed the questionnaire at payment counter and instructed to the employee that take customer consent for the participation in survey and then handover the questionnaire to customer with request to return in next possible visit. The period for data collection was from September 2021 to December 2021. Furthermore, the questionnaire has comprised of questions about each variable and a brief description about objective of survey. Name of the participants was optional.

Sample of the study is very significant for generalizing the outcome of study. The current study has set the sample size using Uma Sekaran Table method and the sample size was 385. Besides, various scholars have suggested different sample sizes like; according to [79] recommended that in marketing studies sample size of 200–500 are valid. In continuation with some of the scholars have suggested that 50 is poor, 300 is good, 500 is very good and 1000 is an excellent sample size [80]. To achieve the minimum sample standard distributed questionnaires were 1000.

**Table 1 ijerph-19-15098-t001:** Measurement Items.

Variables	Authors	Items
Inefficacy	[81]	Most food products’ labels are not clear, so I cannot purchase them
		Most food products need specific proficiency therefore, I avoid these food products
		It is difficult to identify food products that have complex labels.
		I do not trust on the crowded food product labels.
		To read label information I need technical proficiency.
Efficacy	[81]	Easy to read label information is necessary for the right choice of nutritional food.
		It is compulsory to provide information which explain ethical dimension of packaged food
		Packaged food label must be environmental friendly.
		Food processing companies should adhere national rules and regulations for food packaging and ingredients.
Intention	[81]	I have intend to purchase a processed food product
		I have plan to purchase processed food products
		I am willing to purchase processed food products
Attitude towards Familiar food label	[82]	The detail given at familiar food label guides individual at the time of shopping and for me it is significant.
		The available information on familiar labels is appropriate for the selection of healthy processed food and for me it is important.
		A familiar label is an appropriate source for healthy processed food selection and for me it is important.
		Familiar food label is easy to understand and supportive of healthy package food selection.
Attitude towards unfamiliar food label	[82]	The unfamiliar label is not useful for nutritional food selection.
		The available information at unfamiliar labels is difficult to understand
		At unfamiliar food label individual find difficulty to search relevant information for the selection of healthy package food.
		Unfamiliar food label is difficult to understand and support for healthy package food selection.

Structural equation modeling (SEM) has become a popular analysis tool to test structural relationships instead of first-generation analysis. SEM is second generation multivariate analysis tool having several advantages in terms of convenience, efficiency, and accuracy [83,84]. The study used PLS-SEM because it has several advantages over CB-SEM for advanced research analysis [85]. PLS-SEM often preferred because it overcomes the issue of normality and outliers. PLS-SEM considered a silver bullet and holy grails due to its ability to deal with complicated relationships simultaneously [86]. Moreover, PLS-SEM heavily contributes towards exploratory studies and predicts better accuracy [87].

According to [86,88], PLS-SEM analysis the data with two stages approach, i.e., measurement model evaluation and structural model evaluation. Assessment of measurement model involves evaluation of constructs while the structure model involves evaluation of relationship analysis.

Assessment of measurement model has checked through construct reliability and validity, which involve outer loading, composite reliability, average variance extracted, and discriminant validity [89]. The superiority of the model has evaluated through the recommended value of the above mention tests. For the confirmation of convergent validity, the items loading must be greater than 0.5 [90], AVE must be higher than the recommended value of 0.5 and composite reliability greater than 0.7 [91].

There are two criteria to check discriminant validity, i.e., Heterotrait-Monotrait (HTMT) ratio and Fornell–Larcker criterion. The study used Fornell–Larcker criterion to assess discriminant validity [92]. The squared root value of AVE’s (diagonal value) should be higher than the correlation between the latent construct (off-diagonal values).

The structural model evaluates the relationships of latent variables and observed variables. This evaluation has tested through series of assessments such as path coefficients (β), t-values, significance value (p-value), coefficient of determination (R2), effect size (F2), and predictive relevance (Q2). The bootstrapping method (5000 resample) was used for hypotheses testing

## 4. Results and Discussion

Authors of the study have received 831 out of 1000 distributed questionnaires. In primary screening, usable questionnaires were 797 because some participants have filled 50% questionnaires, which excluded from the final analysis. So, 797 sample sizes considered for the studies. The detail presented in Table 2 with the title respondent profile. The data has collected at the time when individuals are shopping, because it gives better idea and opinion about the food selection intention. Additionally, researchers have obtained formal permissions to store before data collection procedure. Each store manager allocated one person as an assistant for data collection. Proper guidance provided to participants as per their demand.

Moreover, the data collected at the point of purchase with the permission of shoppers therefore, the response rate was quite high. Additionally, majority of the respondents lies within the age group of 22 to 30. It indicates that young customer were more interested to participate in food related survey because the popularity of package food products is high as compare to mature population. Besides, the total population of Pakistan is 250 million. In total population the highest representation, which is 64%, belong to age group 22 to 35 [84]. Therefore, the study results can generalize.

## 5. Evaluation of Measurement Model

For the assessment of model authors have used various validity and reliability test. Table 3 and Figure 2 show that all values of outer loading are more significant than the recommended value of 0.5, while values for composite reliability also meet the threshold value of 0.7 ranging from 0.826 (efficacy of food label) to 0.852 (attitude toward unfamiliar food). Similarly, the AVE values are also well above the cut-off value of 0.5, ranging from 0.545 (efficacy of food label) to 0.655 (Healthy packaged food intention).

Table 4 shows that all diagonal values are higher than off-diagonal values, which means discriminant validly is confirmed.

## 6. Structural Model Evaluation

The results indicated that efficacy of food labels (β = 0.507, t = 14.269 > 1.64, *p* < 0.05) and inefficacy of food labels (β = 0.237, t = 6.554 > 1.64, *p* < 0.05) positively and significantly affect the attitude of the consumer towards familiar food labels. Efficacy of food labels (β = 0.236, t = 5.040 > 1.64, *p* < 0.05) found to have significant effect on attitude of the consumer towards unfamiliar food labels while inefficacy of food label (β = 0.049, t = 1.296< 1.64, *p* > 0.05) found to have insignificant effect on attitude of the consumer towards unfamiliar food labels. Moreover, consumers’ attitude of familiar food (β = 0.529, t = 16.015 > 1.64, *p* < 0.05) and unfamiliar food (β = 0.193, t = 5.218 > 1.64, *p* < 0.05) also positively and significantly affect consumer intentions regarding healthy food packages. The R2 value of the model is 0.411, which means the model explains 41% variation in the intention of the consumer regarding healthy food packages. [93] suggested that any value for R2 greater than 0.35 is substantial for any model. So our model is considered as substantial (0.414) as recommended by [93]. Moreover, [93] indicated that value of f2 with 0.02 is poor, with 0.15 is medium, and with 0.35 is strong. The detail has presented in Table 5 and Figure 3. It is argued that if Q2 > 0, the model has predictive relevance [94]. Therefore, the value of Q2 is higher than the rule of thumb which is Q2 = 0.266 and it indicates that the predicted value is high.

### Mediation of Attitude of Familiar and Unfamiliar Food

The study checked the mediation of attitude of the consumer regarding familiar and unfamiliar food between efficacy and inefficacy of food labels and consumer intention for health packaged goods. The results suggested that consumers attitude regarding familiar food mediates (β = 0.276, t = 9.086 > 1.64, *p* < 0.05) the relationship of efficacy of food labels and consumer intention for healthy foods. Finding also confirmed the mediation (β = 0.124, t = 6.2.2 > 1.64, *p* < 0.05) of consumer attitude of familiar food between inefficacy of food labels and consumer intention regarding healthy food packages. As hypothesize, consumer attitude regarding unfamiliar food mediates (β = 0.056, t = 3.338 > 1.64, *p* < 0.05) the relationship of efficacy of food labels and consumer intention of healthy packaged goods. Moreover, the results indicate no mediation of consumer attitude of unfamiliar food (β = 0.014, t = 1.288 < 1.64, *p* > 0.05) between inefficacy of food labels and consumer intention for healthy foods (see Table 6).

## 7. Discussion

The purpose of the present study is to investigate the impact of food label efficacy and inefficacy to make consumer attitude towards both familiar and unfamiliar food labels. Empirical results indicate that efficacy of food label information make consumer attitude even for unfamiliar food labels. On contrary, the inefficacy of food label information has insignificant effect on unfamiliar food label. Moreover, the statistics of present study have unfolded the fact that food label efficacy has pivotal for making an attitude of consumer to consult label information at point of purchase. The efficacy also grabs consumer attention for the products, which they select first time for their shopping. The proposed model comprises of five variables. The researchers have examined how food label efficacy and inefficacy influence in making consumer attitude with familiar and unfamiliar food labels that ultimately leads to strong intention in choosing healthy packaged food. However, to guide consumers about nutrients, food label efficacy and inefficacy is significant for informed decision. The conscious consumer demands food safety. Therefore, food label information has used for informed decision at point of purchase. The most common observation of researchers is the existence of diversity in the food label. The food label is a navigator for consumers for guiding them to the right selection of food but in reality, it is becoming the cause of confusion. Multiple factors affect consumer’s well-being, healthy food is one of them. The increasing trend of packaged food has transformed disease patterns from acute diseases to chronic diseases. To make healthy food selection decisions it is necessary to educate consumers. Furthermore, in the absence of formal criteria regarding the awareness of consumer food label information is an appropriate method. The information is printed at food labels enhance consumers’ awareness about food nutrients like fats, saturated fats, sodium, salt, and fiber. All nutrients are necessary for consumer’s well-being but their balance intake is more essential for a healthy life.

The results indicate that consumers prefer label efficacy in both familiar and unfamiliar food label. The results of present study have linked with previous studies, which suggest that consumers most often judge food products from the labeled information [95]. Reciprocally, the inefficacy and unfamiliarity of food labels create hindrance for consumers while shopping food items. Most often the unfamiliar and complicated food labels create uncertainty in consumer’s minds for nutritious packaged food [96]. Consumers, most often consult food label information to get the insights about composition of these products. The products that are seldom consumed need more information to shape the favorable attitude of an individual. The inefficacy of printed information creates a hindrance to this process. Therefore, the efficacy of food label plays a vital role in developing a positive attitude of consumers to consult label information and develop favorable intentions which are often transformed into actual purchase decisions [97].

Furthermore, consumers have found comfortable with familiar label even the understanding of information is uneasy. Similar results have been examined in past studies and researcher has reported that the food label information is familiar and label format is found to be difficult to understand nevertheless consumer select healthy food items [69]. In contrast, when the food label is unfamiliar and food label information is also difficult to comprehend consumers avoid consulting such products at the point of purchase [98]. Not with standing, regular purchase and consumption make consumers get familiar with the utility of food products. Additionally, while purchasing regularly consumed items, consumers most often do not consult food labels and put them directly into the shopping carts. Though, in regularly consumed edible items, the efficacy or inefficacy of food label information with familiar and unfamiliar food labels, consumers could not make any difference at the point of purchase. As a marketing tactic, if the manufacturer changes the existing format of food labels and the format is new for the consumers still such consumers do not have any concern at the point of purchases [99].

## 8. Theoretical Contribution

The proposed model of the current study is supported by two theories, TRA and PMT. The PMT investigates the factors, which cause the threats and coping of the threats. The involvement of PMT to examine consumer’s nutritional food selection intention has also identified which factors become the cause of threat to avoid food label inform and which factors can be used for coping of these threats. The inefficacy of food label is the threats for consumer to consult food label information, which can be cope with efficacy of food label. Reciprocally, the inefficacy and efficacy affect consumer attitude in reading or avoiding food label, which later influence consumer intention for the choice of nutritional food items.

## 9. Practical Contribution

The practical contribution in current study has linked with the design of the food label, which encourage consumer for healthy food selection. The familiarity of the food label has related to the easy to understandable information written on label. Besides, most of the packaged food products are in regular used items of majority of consumers. Therefore, due to experience consumer has trust on these packaged food firms. Additionally, the outcome of the study reveals that easy to read food labels are very effective. Reciprocally, the food label is the most effective tool for guiding consumers towards nutritional food intention. Therefore, food-processing companies should focus on designing better, effective, and easy to understand food labels with brief, relevant, and comprehensive information. The technical and overcrowded labels create hindrances for customers while shopping packaged food items. Moreover, the efficacy of label can also be a competitive edge for packaged food firms, because consumer will be more comfortable in purchasing package food products from the companies, which have user-friendly food labels.

## 10. Future Direction

The current study paves the way towards a new avenue of research by the efficacy of food labels in influencing consumers’ purchasing decisions. Nevertheless, owing to time constraints, researchers could not achieve other objectives that have been left open for future researchers. Future studies may employ unstructured questionnaires and adopt in-depth interview techniques for data collection. In future, scholars must have to develop a list of processed items, which have consumed occasionally and regularly. This list can further differentiate consumer’s opinions for food label efficacy and inefficacy at point of purchase. Although the current study is based on the data at the point of purchase nevertheless the structured questionnaires not provide liberty to consumers to express their opinions in detail. Therefore, a qualitative data analysis technique opted in future studies. In addition to, the past studies have claimed that strong intention transformed into actual behavior. Therefore, future researchers should involve actual behavior in the proposed model.

## 11. Conclusions

This research aimed to examine consumers’ intention to nutritional processed food selection. Owing to the absence of proper guidance the role of label is inevitable. Food label information is mandatory but most often consumers need special proficiency and intention to understand food label information. The current study proposed a model and suggested that food label efficacy is significant for informed purchase decision.

## Figures and Tables

**Figure 1 ijerph-19-15098-f001:**
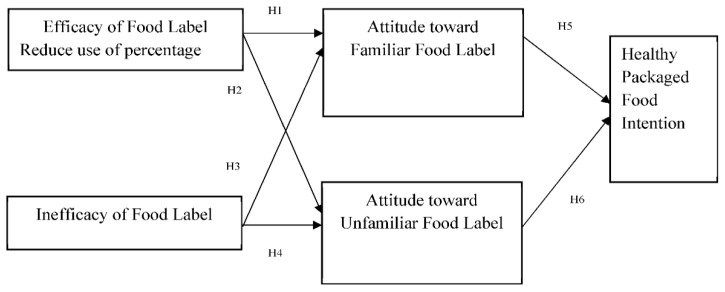
Food labeling and consumer intention for healthy package food.

**Figure 2 ijerph-19-15098-f002:**
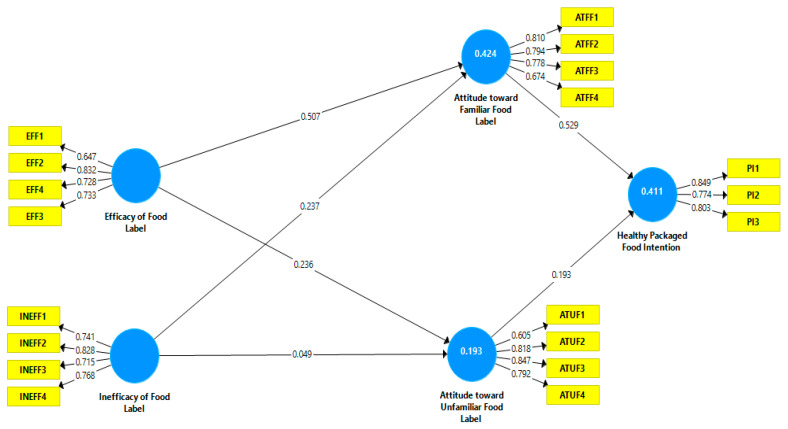
Outer loadings.

**Figure 3 ijerph-19-15098-f003:**
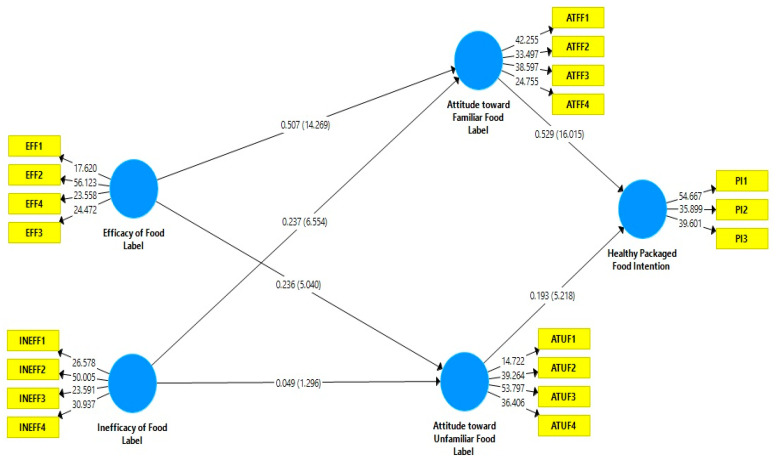
Hypothesis testing (Bootstrapping results).

**Table 2 ijerph-19-15098-t002:** Respondent profile.

	Characteristics	Frequency	Percentage
Gender	Male	503	63.11
	Female	294	36.89
Age	18–21	241	30.24
	22–25	289	36.26
	26–29	180	22.58
	30-over	87	10.92
Education Level	Undergraduate	356	44.67
	Graduate	305	38.27
	Postgraduate	136	17.06
Shopping Method	Shopping Single	355	44.54
	Shopping with Family with kids	263	32.99
	Shopping with Family without kids	179	22.47

**Table 3 ijerph-19-15098-t003:** Measurement model evaluation.

Constructs	Item	Loading	CR	AVE
Efficacy of food label	EFF1	0.647	0.826	0.545
	EFF2	0.832		
	EFF3	0.728		
	EFF4	0.733		
Inefficacy of food label	INEFF1	O.741	0.848	0.584
	INEFF2	0.828		
	INEFF3	0.715		
	INEFF4	0.768		
Attitude toward Familiar food	ATFF1	0.810	0.850	0.586
	ATFF2	0.794		
	ATFF3	0.778		
	ATFF4	0.674		
Attitude toward Unfamiliar food	ATUF1	0.605	0.853	0.595
	ATUF2	0.818		
	ATUF3	0.847		
	ATUF4	0.792		
Healthy Packaged food Intention	PI1	0.850	0.850	0.655
	PI2	0.773		
	PI3	0.804		

**Table 4 ijerph-19-15098-t004:** Discriminant validity (Fornell–Larcker criterion).

	ATTF	ATTUF	EFF	PI	INEFF
ATTF	0.766				
ATTUF	0.458	0.771			
EFF	0.616	0.364	0.738		
PI	0.618	0.435	0.685	0.809	
INEFF	0.470	0.387	0.459	0.406	0.764

**Table 5 ijerph-19-15098-t005:** Hypothesis testing (structure model results).

	Path Coefficients	SD	t-Value	*p*-Value	Supported	R^2^	Q^2^	F^2^
EFF → ATFF	0.507	0.035	14.269	0.000	Yes	0.414	0.266	0.380
INEFF → ATFF	0.237	0.037	6.554	0.000	Yes			0.076
EFF → ATUF	0.236	0.048	5.040	0.000	Yes			0.082
INEFF → ATUF	0.049	0.047	1.296	0.183	No			0.003
ATFF → PI	0.529	0.037	16.015	0.000	Yes			0.383
ATTUF → PI	0.193	0.038	5.218	0.000	Yes			0.049

EFF (Efficacy of food-label), INEFF (In-Efficacy of Food-Label), ATFF (Attitude towards Familiar Food-Label), ATUF (Attitude towards Unfamiliar Food-Label) and PI (Purchase Intention).

**Table 6 ijerph-19-15098-t006:** Mediation results.

	Path Coefficients	SD	t-Value	*p*-Value	Supported
EFF → ATFF → PI	0.276	0.030	9.086	0.000	yes
INEFF → ATFF → PI	0.124	0.020	6.202	0.000	yes
EFF → ATUF → PI	0.056	0.017	3.338	0.000	yes
INEFF → ATUF → PI	0.014	0.011	1.288	0.198	No

## Data Availability

The primary data used for the analysis of the current study. The data collected through adapted questionnaire with the consent of respondents. All respondents have shown their consent to participate in the research.

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
