# Peer review of "The Impact of Interpretive Packaged Food Labels on Consumer Purchase Intention: The Comparative Analysis of Efficacy and Inefficiency of Food Labels"

_ijerph, 2022, doi:10.3390/ijerph192215098_

Round 1

Reviewer 1 Report

I found the manuscript very interesting for the topic dealt with. The article also provides useful and documented information on consumer behaviour.  It is well written but needs a smooth rebalancing between sections.

Introduction. The introduction seems too long and sometimes redundant in explaining the elements considered. I appreciated the paragraph structure but the whole section should be overally better addressed in order to make it more reader friendly. Also, I suggest using part of the references to strengthen the discussion section (if possible).

Materials and methods. Expand the methods as they only examine how the survey is conducted but not the methodology for processing the data. It is suggested that the methods would be expanded with details on how the questionnaire responses are used to calculate the parameters and how the variables have been treated to achieve the results. Parts of the results are more related to M&M.

Results are ok but some parts (those highlighted in the attached document) are to be summarized here and reported in depth in the methods section.

Discussion. As mentioned in the comment to the introduction section, the discussion should be rebalanced trying to use more references (if possible).

Conclusions. The conculsions are rather short. I would suggest to reorganizing the section using paragraphs 9, 10, 11, 12 in order to specify the implications of the study and research findings.

References. I recommend that you unify the references along the text. For example: (Giacalone and Jaeger, 2019)[82] or just [82].

Furthermore, some references highlighted in the attached documents are missing or seems not relevant.

Author Response

Response to Reviewer 1 Comments

Point 1: Introduction. The introduction seems too long and sometimes redundant in explaining the elements considered. I appreciated the paragraph structure but the whole section should be overally better addressed in order to make it more reader friendly. Also, I suggest using part of the references to strengthen the discussion section (if possible). 

Response 1: We are very thankful for your suggestion. We have written the introductory part in detail for more clarity and understanding of the reader. Nevertheless, authors have removed the unnecessary sentences which have created redundancy. The changes are incorporated at page no 2. 

Point 2: Materials and methods. Expand the methods as they only examine how the survey is conducted but not the methodology for processing the data. It is suggested that the methods would be expanded with details on how the questionnaire responses are used to calculate the parameters and how the variables have been treated to achieve the results. Parts of the results are more related to M&M.

Response 2: The changes are incorporated at page no. 8 with track changes.

Point 3: Results are ok but some parts (those highlighted in the attached document) are to be summarized here and reported in depth in the methods section.

Response 3: The changes are incorporated and reported at page no. 9 with track changes.

Point 4: Discussion. As mentioned in the comment to the introduction section, the discussion should be rebalanced trying to use more references (if possible).

Response 4: Changes are incorporated at page no. 13.

Point 5: Conclusions. The conculsions are rather short. I would suggest to reorganizing the section using paragraphs 9, 10, 11, 12 in order to specify the implications of the study and research findings.

Response 5: Changes are incorporated at page no. 14 and highlighted with black and bold.

Point 6: References. I recommend that you unify the references along the text. For example: (Giacalone and Jaeger, 2019) [82] or just [82].

Response 6: All changes are incorporated as per the suggestion.

Reviewer 2 Report

This study investigated the title " The Impact of Interpretive Packaged Food Label on Consumer Purchase Intention: The Comparative Analysis of Efficacy and Inefficiency Food Label" After revision, I think it is an interesting article.

  1. The Abstract points out that "the data of 797 respondents have been collected from three big grocery stores", but in the Methods point, "the study selected four retail outlets." Please check it.
  2. P8, the title "3. Results and Discussion" should be "4. Results and Discussion", Please check it. The same, there are no 6. please check it.
  3. P10, 7. Structural Model Evaluation, suggestion to delete because the paragraph does not help the reader to read. 
  4. P9-10 , Table 3. and Figure 2. What are EFF1-4, INEFF1-4, ATFF1-4, ATUF1-4, and PI1-3 in the item? Please note.
  5. In addition, married and having children are important factors in the purchase. Did the author consider these two variables?
  6. Please change the description to the third person and avoid using "our or we" (Just a suggestion).

Author Response

Response to Reviewer 2 Comments

Point 1: The Abstract points out that "the data of 797 respondents have been collected from three big grocery stores", but in the Methods point, "the study selected four retail outlets." Please check it.

Response 1: The change has incorporated with tack change method. It is a typo error. The data collected from four big retail stores.

Point 2: P8, the title "3. Results and Discussion" should be "4. Results and Discussion", Please check it. The same, there are no 6. please check it.

Response 2: The changes are incorporated with track changes method.

Point 3: P10, 7. Structural Model Evaluation, suggestion to delete because the paragraph does not help the reader to read.

Response 3: The changes are incorporated with track changes method.

Point 4: P9-10, Table 3. and Figure 2. What are EFF1-4, INEFF1-4, ATFF1-4, ATUF1-4, and PI1-3 in the item? Please note.

Response 4: Changes are incorporated and the complete names with abbreviation are written under the table for better understanding of reader.

Point 5: In addition, married and having children are important factors in the purchase. Did the author consider these two variables?

Response 5: In current study objective of the authors was not bring the population variable as a control variable. Therefore, authors have not conducted a test which can analyse the opinions of married and with children consumers.  

Point 6: Please change the description to the third person and avoid using "our or we" (Just a suggestion).

Response 6: The suggested changes are incorporated at page no 5 with tack changes.